# Carbon Fiber-Reinforced PolyEtherEtherKetone (CFR-PEEK) Instrumentation in Degenerative Disease of Lumbar Spine: A Pilot Study

**DOI:** 10.3390/bioengineering10070872

**Published:** 2023-07-23

**Authors:** Riccardo Ghermandi, Giovanni Tosini, Alberto Lorenzi, Cristiana Griffoni, Luigi La Barbera, Marco Girolami, Valerio Pipola, Giovanni Barbanti Brodano, Stefano Bandiera, Silvia Terzi, Giuseppe Tedesco, Gisberto Evangelisti, Annalisa Monetta, Luigi Emanuele Noli, Luigi Falzetti, Alessandro Gasbarrini

**Affiliations:** 1Department of Spine Surgery, IRCCS Istituto Ortopedico Rizzoli, 40136 Bologna, Italy; riccardo.ghermandi@ior.it (R.G.); marco.girolami@ior.it (M.G.); valerio.pipola@ior.it (V.P.); giovanni@barbantibrodano.com (G.B.B.); stefano.bandiera@ior.it (S.B.); silvia.terzi@ior.it (S.T.); giuseppe.tedesco@ior.it (G.T.); gisberto.evangelisti@ior.it (G.E.); annalisa.monetta@ior.it (A.M.); luigi.falzetti@ior.it (L.F.); alessandro.gasbarrini@ior.it (A.G.); 2Ortopedia e Traumatologia 3 ad Indirizzo Chirurgia Vertebrale, AOU Città della Salute e della Scienza (Presidio CTO), 10126 Torino, Italy; alberto.lorenzi@edu.unito.it; 3Laboratory of Biological Structure Mechanics, Department of Chemistry, Materials and Chemical Engineering “G. Natta”, Politecnico di Milano, 20133 Milano, Italy; luigi.labarbera@polimi.it; 4IRCCS Istituto Ortopedico Galeazzi, 20157 Milano, Italy

**Keywords:** degenerative lumbar spine disease, carbon fiber, CRF-PEEK, spine fusion, spondylolisthesis

## Abstract

CFR-PEEK is gaining popularity in spinal oncological applications due to its reduction of imaging artifacts and radiation scattering compared with titanium, which allows for better oncological follow-up and efficacy of radiotherapy. We evaluated the use of these materials for the treatment of lumbar degenerative diseases (DDs) and considered the biomechanical potential of the carbon fiber in relation to its modulus of elasticity being similar to that of bone. Twenty-eight patients with DDs were treated using CRF-PEEK instrumentation. The clinical and radiographic outcomes were collected at a 12-month FU. Spinal fusion was evaluated in the CT scans using Brantigan scores, while the clinical outcomes were evaluated using VAS, SF-12, and EQ-5D scores. Out of the patients evaluated at the 12-month FU, 89% showed complete or almost certain fusion (Brantigan score D and E) and presented a significant improvement in all clinical parameters; the patients also presented VAS scores ranging from 6.81 ± 2.01 to 0.85 ± 1.32, EQ-5D scores ranging from 53.4 ± 19.3 to 85.0 ± 13.7, SF-12 physical component scores (PCSs) ranging from 29.35 ± 7.04 to 51.36 ± 9.75, and SF-12 mental component scores (MCSs) ranging from 39.89 ± 11.70 to 53.24 ± 9.24. No mechanical complications related to the implant were detected, and the patients reported a better tolerance of the instrumentation compared with titanium. No other series of patients affected by DD that was stabilized using carbon fiber implants have been reported in the literature. The results of this pilot study indicate the efficacy and safety of these implants and support their use also for spinal degenerative diseases.

## 1. Introduction

Over the last few decades, metal implants have become commonly used in spinal surgery for the stabilization of the spinal column using pedicle screws and rods to treat different pathologies of the spine; these types of instrumentation have dramatically increased the union rate in spinal fusion. Spinal implants need to have good performance in biofunctionality and biocompatibility. Biofunctionality is concerned with the mechanical properties of alloys, as high yield strength, stiffness, and fatigue are required. Biocompatibility refers to the interaction of an alloy with the internal environment of the human body; the integration of the implant with bone is relevant to reduce the possibility of device failure or loosening. Due to its remarkable mechanical strength, high fatigue/corrosion resistance, low density, high biocompatibility, and possibility to be contoured intraoperatively for a tailored fit, titanium has become the metal of choice for spine surgery, replacing stainless steel implants. Titanium and titanium alloys are also used in orthopedic surgery for components of prostheses in total hip arthroplasty, total knee arthroplasty, and fracture fixation. Moreover, it is possible to enhance titanium’s properties by combining it with other elements (i.e., Zr, O, Al) or by modifying its manufacturing process. However, metal implants also have disadvantages, which include limited fatigue life, low wear resistance, a mismatch of modulus elasticity between the metal and the bone, potential for debris generation, allergic host response, and imaging artifacts in MRI, X-rays, and CT evaluations [1,2,3].

Over the last few years, glass fibers (GFs) and carbon fibers (CFs) have been largely studied and used in the production of advanced polymer composites, i.e., glass fiber-reinforced polymer (GFRP) and carbon fiber-reinforced polymer (CRFP) composites. In particular, the main characteristics of CFs, such as their light weight, high stiffness, high tensile strength, good fatigue resistance, good vibration damping, high-temperature stability, high chemical/corrosion endurance, good electro-magnetic properties, electrical conductivity, X-ray permeability, and low coefficient of thermal expansion, have made them very popular in various engineering applications, including aerospace, automobile and marine transport, military, civil engineering, sporting goods, and medical application (surgery and X-ray equipment, prostheses, tendon/ligament implants) [4]. Recently, carbon fibers have been mixed with synthetic and natural fibers to obtain reinforced composite materials in different areas [5].

Carbon fiber-reinforced (CRF) composite materials have also been developed to help overcome some of the disadvantages of standard metallic implants. Their mechanical properties depend on the carbon fibers’ amount, alignment, and length, and they are tuned to match the qualities of human bone [6]. In particular, CFR-PEEK has been used to realize many different types of orthopedic and spinal implants (e.g., plates, nails, pedicle screws, spinal rods), and it has been gaining popularity in the spine and appendicular skeleton oncologic applications because of the reduction of artifacts in imaging procedures (MRI and CT) compared with titanium, allowing for more feasible tumor follow up, simpler radiotherapy planning, and potentially greater safety and quality for radiation therapy [7,8].

The use of CFR-PEEK instrumentation (rods and pedicle screws) for the treatment of primary and metastatic spinal tumors has been recently described in several clinical studies [9,10,11,12], and it has been demonstrated that postoperative CT and MRI scans show reduced artifacts in patients who received CFR-PEEK pedicle screws for the treatment of spinal tumors when compared with standard titanium alloy implants [13].

Concerning the biomechanical literature, CFR-PEEK rods have been shown to have tunable mechanical properties, particularly with respect to the bending and compressive moduli and strength to parameters that are comparable or superior to those of cortical bone [14]. CFR-PEEK can be used for semirigid stabilization, and it demonstrates bending stiffness, yield and ultimate loads, and a fatigue strength that is comparable with or higher than rigid titanium systems [15]; although the torsional stiffness and yield torque are lower than titanium implants, they are suitable for withstanding the physiological loads. Moreover, CFR-PEEK rods can provide effective primary stability, better kinematics, and improved load-sharing compared with titanium implants [16]. CFR-PEEK pedicle screws have been shown to have higher pullout strength [15] and comparable or higher resistance to screw loosening than traditional implants [17,18], which is explained by the absence of the mismatch between the mechanical properties of CFR-PEEK and the surrounding bone, thus reducing the stresses and strains at the bone-implant interface. CFR-PEEK also has good biocompatibility, enhanced osteogenic property, and bioactivity [19,20]. Radiolucency and CT/MRI artifact-free features may offer additional advantages over traditional spine fixation systems that are constructed using metal alloys [21]. These biomechanical properties have prompted us to think that the use of CRF-PEEK could also be advantageous for the stabilization of the spine in patients affected by degenerative diseases.

Recently, a study investigated the use of CFR-PEEK instrumentation for the treatment of degenerative spinal disorders [22]. The authors analyzed artifacts formation at MRI assessment and compared titanium and CRF-PEEK instrumentation. CRF-PEEK pedicle screws exhibited smaller artifact areas on vertebral body surfaces and their surrounding tissues, improving the radiographic assessability and providing a diagnostic benefit.

A few biomechanical comparative studies have demonstrated the potential of CFR-PEEK for enhancing the load-sharing on the anterior column and reducing the loads on the adjacent proximal disc [23,24], but no clinical study has been published yet. This paper presents a case series of patients treated with CRF-PEEK instrumentation for the posterior stabilization of the spine in degenerative lumbar spine disease with the purpose of assessing the fusion rate and clinical outcomes.

## 2. Materials and Methods

Patients who underwent lumbar interbody fusion with CFR-PEEK instrumentation in our institution from October 2015 to November 2021 were enrolled in two prospective clinical studies, which was approved by the local ethics committee on October 2014 (protocol number 0033475) and on July 2018 (protocol number CE-AVEC: 208/2018/Disp/IOR). In the first study, the CarboClear^®^ system (CarboFix Orthopedics, Herzliya, Israel) was used, while in the second study, the BlackArmor^®^ system (Icotec AG, Altstaetten, Switzerland) was used. Both implants had similar characteristics, with polyaxial screws and rods that were preformed in lordosis and had a 5.5–6.0 mm diameter. Patients were included in the study following the signature of a study-specific consent form. Inclusion criteria were patients aged 18 years or more that were affected by degenerative lumbar disease or low-grade spondylolisthesis. The exclusion criteria were high-grade spondylolisthesis, oncologic or infectious disease, previous spinal instrumentation, bone metabolic disease or use of osteoanabolic drugs, and obesity.

The consecutive steps of the study are reported in a flow chart (Figure 1).

All of the patients enrolled received a standard open PLIF/TLIF procedure with a composite CFR-PEEK fixation system (CarboClear or BlackArmor) and combined PEEK core/titanium-surfaced interbody cage (Concorde^®^ ProTi 360° System, Depuy Synthes, Palm Beach Garden, FL, USA, with the dimensions being 9 × 23 or 9 × 27 mm and the height ranging from 7 to 15 mm). The spinal instrumentation with pedicle screws and rods is represented in Figure 2. While screws and rods provide primary stability to the spine, a bone graft was added to achieve a stable spinal fusion.

Post-op CT scans and standard radiograms before discharge were collected. Patients were evaluated at 6 and 12 months. Patient-reported outcomes (VAS, SF-12, and EQ-5D scores) were also collected at baseline and at a 12-month follow up. The authors asked the patients if they felt the presence of the spinal instrumentation and if this presence was particularly relevant in association with a change in weather or temperature. The possible answers to the question: “when you feel the presence of the implants in your spine?” were scored as follows: 1 = never; 2 = only during weather/temperature change; 3 = occasionally; 4 = always. Adverse events were collected in the intra-operative, post-operative, and follow up periods and were classified according to SAVES v2 [25].

Radiographic images and a CT scan at 12 months were used to determine the degree of fusion and bone regeneration. Bone regeneration and the degree of fusion were determined by an independent radiologist using a CT scan analysis. Bone regeneration was identified as the presence of a continuous trabecular bone bridge along with the lack of radiolucency, as assessed by the diagnostic imaging (CT scan), and it was evaluated by using Brantigan–Steffee classification, which assesses spinal fusion from grade A (pseudoarthrosis) to grade E (certain fusion) [26]. Spinal fusion was considered successful if the scores were D and E.

### Statistical Analysis

No sample size calculation was performed because of the study’s design (pilot study). Considering the small number of patients treated, no specific statistical analyses have been carried out. A descriptive statistical analysis has been provided for the clinical scores (VAS, ODI, and EQ-5D) and for the fusion assessment.

Results are presented as the number (n), mean ± standard deviation, and percentage, as appropriate. After having verified a normal distribution and homogeneity of variance, a two-way ANOVA test was performed to detect the changes from baseline to the follow-up scores. The level of statistical significance was set at *p* < 0.05. GraphPad Prism software was used.

## 3. Results

Twenty-eight patients were enrolled following the informed consent approval and signature. The demographic and clinical data are reported in Table 1. All of the patients had a diagnosis of disc degenerative disease and underwent a transforaminal lumbar interbody fusion (TLIF) procedure with CRF-PEEK rods, pedicle screws, and carbon fiber cages. In three cases, the procedure was performed using minimally invasive surgery. Nineteen patients (67.9%) had only one instrumented level, and the most frequently treated level was L5 (41.8%). Six patients (21.4%) were previously surgically treated without instrumentation (herniectomy or interspinous device). The mean follow-up period was 43 months. One intra-operative adverse event was observed in the form of L5 screw malpositioning; no other adverse events were collected.

Spinal fusion was assessed in the CT scan examinations performed 12 months after surgery and was evaluated using the Brantigan–Steffee classification, as reported in Table 2. A total of 24 cases (89% of patients) had the radiographic fusion scored as D or E, thus being considered successful.

Moreover, at 12 months follow-up, all clinical outcomes improved as reported in Figure 1. The VAS score decreased from 6.81 ± 2.01 to 0.85 ± 1.32 (Figure 1), the EQ-5D score increased from 53.4 ± 19.3 to 85.0 ± 13.7 (Figure 1), the SF-12 physical component score (PCS) increased from 29.35 ± 7.04 to 51.36 ± 9.75 (Figure 1), and the SF-12 mental component score (MCS) increased from 39.89 ± 11.70 to 53.24 ± 9.24 (Figure 3).

Nine patients had the degeneration grade of the proximal adjacent intervertebral disc evaluated using MRI before and after surgery by the Pfirrmann score. In eight cases, no change was observed in the Pfirrmann grade, whereas in two cases, the Pfirrmann grade increased.

Finally, we asked patients to report about their perception of the presence of spinal instrumentation, particularly in correlation with the weather and changes in temperature. The results of the survery are reported in Table 3. A total of 71% of patients never felt the presence of spinal instrumentation and 21% of patients only felt discomfort when there were changes in weather and temperature. Most patients with titanium instrumentation complain of some discomfort due to its presence, which is increased when there are climatic changes.

In Figure 4 and Figure 5, we reported radiographic imaging concerning two cases of patients affected by DD that were treated with CRF-PEEK posterior instrumentation and TLIF. In the first case, the levels treated were L4–S1, and in second case, they were L5–S1. In both cases, we can observe the presence of a relevant posterior fusion (Figure 4 and Figure 5, panels D) and interbody fusion (Figure 4 and Figure 5, panels E) at 12 months follow-up.

## 4. Discussion

CFR-PEEK is an innovative composite material combining the deformability of PEEK with the strength of carbon fibers, and it offers several advantages compared with the standard materials (i.e., Ti6Al4V alloys) used in the spine field. Its mechanical properties depend on the carbon fibers’ amount, alignment, and length, and they are typically tuned to match the properties of human bone [6]. The bending and compressive moduli and the strength are comparable or even superior to that of cortical bone [14], but the bending stiffness, yield and ultimate loads, and fatigue strength were found to be comparable with or higher than traditional titanium alloys [15], with the torsional stiffness and yield torque being suitable to withstand the physiological loads. The additional advantages for using CFR-PEEK in the spine field relies on good biocompatibility, an enhanced osteogenic property, and bioactivity [19,20], while radiolucency and CT/MRI artifact-free features may offer additional advantages over traditional metallic systems, both for routine imaging and radiotherapy treatments. From an imaging perspective, CRF-PEEK instrumentation allows for an improved evaluation of adjacent anatomic structures during radiography, CT, and MRI scans. This results in improved post-operative surveillance imaging as well as an easier visualization of the anatomy for image-guided percutaneous interventions. In radiation oncology treatment, CRF-PEEK devices are also advantageous due to their decreased imaging artifact during planning and decreased dose perturbation during RT delivery. For these reasons, the manufacturing processes for CRF-PEEK materials continue to evolve and improve, and orthopedic applications for them in the spine and the appendicular skeleton are increasing [8,9].

Previous biomechanical preclinical studies have confirmed the effectiveness and efficacy of CFR-PEEK material for use in interbody fusion and vertebral reconstruction in oncologic patients. Brantigan et al. pioneered the usage of CFR-PEEK PLIF cages and provided comparable compressive stiffness as the intact spine, but higher strength and superior pull-out resistance compared with bone allografts [27]. Disch et al. [28,29] reported that CFR-PEEK vertebral body replacements (VBRs) could provide effective primary stability with long posterior fixation following en bloc spondylectomy, and it had a similar performance to expandable titanium devices [28,29]. The first case series of bone tumors of the spine surgically reconstructed with a new, custom, fully radiolucent, polyetheretherketone/carbon fiber (PEEK/CF) vertebral body replacement (VBR) integrated system has been recently reported [30]. More recently, rods and pedicle screws for spinal stabilization have also been realized using CRF-PEEK material [15]. Previous findings support the osteoconductive properties of CFR-PEEK, which has mechanical properties that are comparable with the surrounding bone, thus ensuring more homogeneous stress and strain distributions at the bone–screw interface, minimizing micromotions, and promoting an enhanced load-sharing on the anterior column. This biomechanical rationale also explains the results of recent clinical studies that support reduced cage subsidence and enhanced osteointegration and fusion.

Boriani et al. [10] studied the safety and efficacy of a CFR-PEEK composite rod and screw fixation system compared with titanium implants. Thirty-four patients who underwent thoracic and lumbar spinal fixation for spinal tumors using CFR-PEEK composite implants were followed for a mean of 13 months. Early follow-up findings revealed neurological improvement and improved pain control. Only one intraoperative screw breakage occurred out of 232 implanted screws, and the loosening of 2 sacral screws was detected at 12 months. The authors reported that six local recurrences were detected early thanks to the implant radiolucency. Additionally, they described a case series of six patients who underwent poster fixation of the cervical spine using a hybrid system containing CFR-PEEK screws and rods with polyester bands and titanium clamps. In this report, no intraoperative complications occurred in any of the patients [12]. Tedesco et al. [11] presented a study on 22 patients who underwent spinal stabilization using a CFR-PEEK system with a median follow-up of 10 months. The authors reported only one intraoperative screw breakage, which occurred in the absence of rod breakage, during the follow-up period. This was supported by Laux et al. [31] who demonstrated that CFR-PEEK implants have no increased risk of perioperative instrumentation complications. Pipola et al. [32] presented a case of a patient who underwent an en bloc resection with an anterior and posterior approach for the treatment of sclerosing epithelioid fibrosarcoma and spinal fixation using composite PEEK-CF rods, and a 2-year follow-up proved the implant’s stability with no local recurrences found. Based on these results, CFR-PEEK possesses a similar efficacy profile to that of the standard titanium instrumentation but has an increased safety profile with respect to the early detection of tumor recurrency.

In addition to evaluating the dosimetric impact caused by CFR-PEEK, Mastella et al. [33] analyzed the CT phantoms of the two screws that were acquired to evaluate image quality. CFR-PEEK screws did not create appreciable artifacts on the CT scans compared with titanium implants, thus improving dose accuracy and image quality. Ringel et al. [13] reported a reduction in screw artifacts in MRI and CT scans for CFR-PEEK implants compared with titanium implants. All of these papers report about the clinical applications of CRF-PEEK systems for the treatment of spinal tumors, and the advantages of radiolucency and artifact-free imaging are very attractive for radiotherapy and treatment monitoring.

Fleege et al. [22] recently compared two groups of patients that were treated for degenerative spinal disorders of the lumbar spine using titanium pedicle screws or CF-PEEK pedicle screws. All patients underwent an MRI assessment within the first four postoperative weeks. CF-PEEK pedicle screws exhibited smaller artifact areas on the vertebral body surfaces and their surrounding tissues due to them having a lower density than titanium (similar to bone), which significantly improved the radiographic assessability.

The osteoconductive properties of CFR-PEEK are considered to also be optimal for pedicle screw anchorage because stress and strain gradients may be minimized at the screw–bone interface, thus improving short-term primary stability and enhancing long-term secondary stability throughout osteointegration. Several preclinical in vitro studies have confirmed this idea; they have reported higher axial pullout strength for CFR-PEEK poly-axial pedicle screws compared with titanium screws [15] and reduced screw loosening when used for the semi-rigid stabilization of osteoporotic spines [18]. Based on these advantageous characteristics, we proposed conducting a pilot study on degenerative lumbar patients that were entirely treated using CFR-PEEK instrumentation.

We analyzed a case series of 28 patients and observed a high rate of spinal fusion (89% of cases), which is comparable with the results reported in the literature for titanium implants [34]. Moreover, no mechanical complications such as screw mobilization, rod breakage, or cage subsidence were recorded during the follow-up period. These encouraging radiographic outcomes were associated with a significant improvement of clinical outcomes, as measured by PROs, including pain reduction and the improvement of the health-related quality of life.

A few biomechanical, comparative, in silico studies [23,24] have studied the usage of CFR-PEEK for semirigid posterior stabilization, and they can be used to interpret the results of the clinical study collected here. The osteoconductive properties of CFR-PEEK promote a better load-sharing on the anterior column and an improved osteointegration compared with more rigid posterior instrumentations.

Comparative biomechanical studies also demonstrate that semi-rigid posterior stabilization using CFR-PEEK could reduce the loads (specifically the intradiscal pressure, facet joint force, and interspinous and supraspinous ligaments elongation) on the adjacent proximal disc compared with stiffer materials, such as titanium alloys and stainless steel [23,24].

According to a meta-analysis, the pooled incidence of ASD after lumbar fusion surgery was 26.6% [35]. In our study, only nine patients had pre-operative and post-operative MR imaging, which allowed for a comparison of the adjacent proximal disc degeneration before and after surgery. In all cases, a pre-existing adjacent segment degeneration (ASD) was detected, and we observed only one case of worsening degeneration following surgery. However, the limited amount of data regarding the characteristics of the adjacent proximal disc does not allow us to attribute a greater efficacy of the carbon implants to preserve the adjacent disc from degeneration compared with titanium.

Moreover, the effect of cold weather has been investigated in patients with orthopedic implants, but this was not related to any specific implant type, material, or site [36]. The results of the survey that we conducted indicates that patients’ sensitivity to the presence of instrumentation during weather changes (i.e., temperature and humidity) is low, which may depend on the intrinsic thermal properties of CFR-PEEK, such as the coefficients of thermal expansion and thermal conductivity, which are in the same range as the parameters reported for human cortical and highly mineralized trabecular bone (Table 4). Although a direct comparison with patients treated with traditional titanium implants is not available, this aspect may require further investigation to establish the specific factors that are involved in patients’ sensitivity to implant material.

## 5. Conclusions

Despite some relevant limitations, due to the small cohort of patients analyzed and the lack of a control group treated using traditional titanium implants, this is the first clinical study supporting the safety and efficacy of implants that are entirely constructed out of CRF-PEEK for spinal stabilization in patients affected by degenerative diseases. Our study demonstrates that the use of CRF-PEEK implants is associated with good clinical and radiographic outcomes, a high fusion rate, the reduction of pain, and the improvement of quality of life. The preliminary data also suggest a preservation of the adjacent proximal disc degeneration and a good tolerability of the instrumentation for the patients. These advantageous characteristics, which are associated with the reduction of imaging artifacts in CT and MRI procedures, support the use of CRF-PEEK implants as a valid alternative to conventional titanium devices for spinal stabilization in degenerative patients.

Further investigations can be performed to evaluate the effect of CRF-PEEK instrumentation on adjacent segments compared with titanium devices and to analyze the tolerability of the material; these aims are important in order to identify possible selection factors for patients who may benefit more from the use of CFR-PEEK implants compared with titanium ones.

## Figures and Tables

**Figure 1 bioengineering-10-00872-f001:**
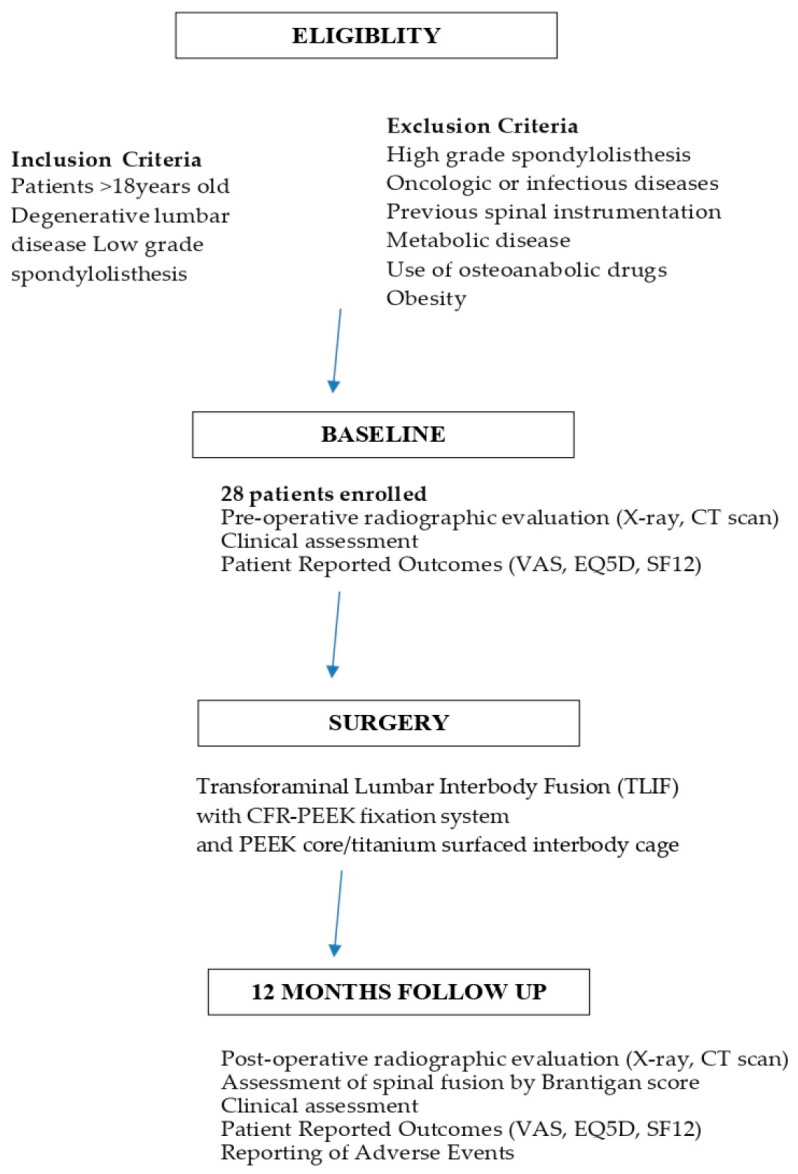
Flow chart of the study.

**Figure 2 bioengineering-10-00872-f002:**
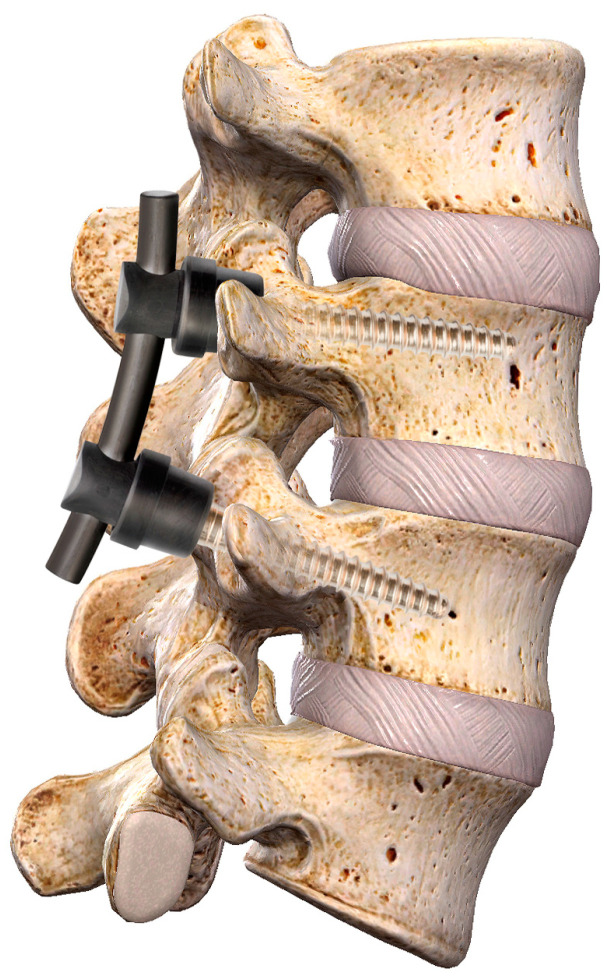
Image of a section of the spine where the pedicle screws and bars connecting the screws have been inserted to stabilize the spine.

**Figure 3 bioengineering-10-00872-f003:**
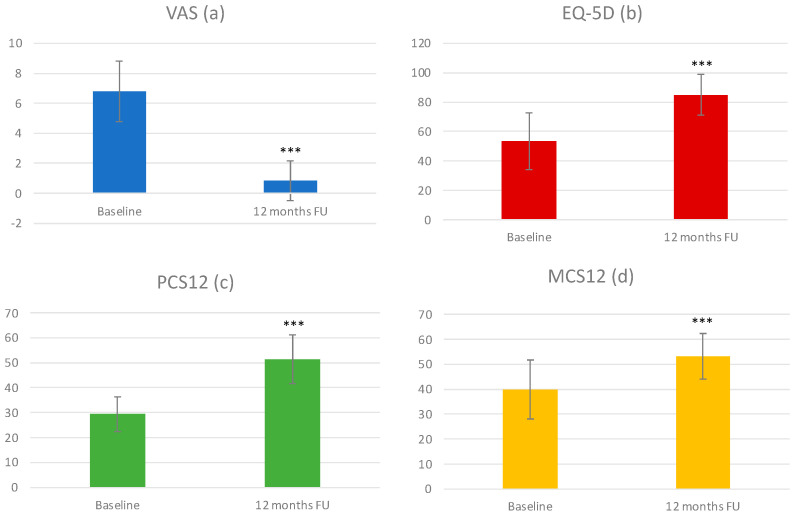
(**a**) Plot of the visual analog scale (VAS) score as evaluated preoperatively and at 12 months follow-up. The black asterisks show a significant difference between post-operative and preoperative values (*** *p* < 0.0005). (**b**) Plot of the EQ-5D score as evaluated preoperatively and at 12 months follow-up. The black asterisks show a significant difference between post-operative and preoperative values (*** *p* < 0.0005). (**c**,**d**) Plot of the SF12 physical component (PCS12) and mental component (MCS12) as evaluated preoperatively and at 12 months follow-up. The black asterisks show a significant difference between post-operative and preoperative values (*** *p* < 0.0005).

**Figure 4 bioengineering-10-00872-f004:**
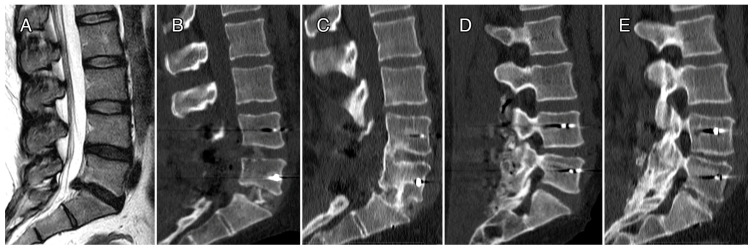
Case 1. Female, 40 yo, DD L4–S1, chronic low back pain and leg pain, TLIF and arthrodesis L4–S1. Pre-op lumbar spine MRI (**A**); post-op CT sagittal view (**B**) and CT at 12 months FU sagittal view (**C**); post-op CT sagittal view (**D**) and CT at 12 months FU sagittal view (**E**). Brantigan score at 12 months FU.

**Figure 5 bioengineering-10-00872-f005:**
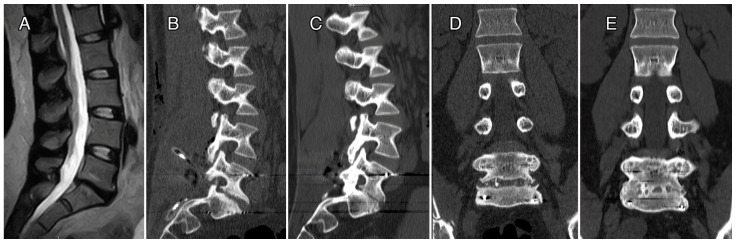
Case 2. Female, 42 yo, DD L5–S1, chronic low back pain, TLIF and arthrodesis L5–S1. Pre-op lumbar spine MRI (**A**); post-op CT sagittal view (**B**) and CT at 12 months FU sagittal view (**C**); post-op CT coronal view (**D**) and CT at 12 months FU coronal view (**E**). Brantigan score at 12 months FU.

**Table 1 bioengineering-10-00872-t001:** Demographic and clinical data.

	Study Sample (Total Number = 28 Patients)
**Age**, year median (range)	48 (20–75)
**Sex**, n (%)	
Male	14 (50.0)
Female	14 (50.0)
**Number of instrumented levels ^a^**, n (%)	
1	19 (67.9)
2	7 (25)
3	2 (7.1)
**Levels instrumented ^b^**, n (%)	
L2	1 (1.5)
L3	3 (4.5)
L4	17 (25.4)
L5	28 (41.8)
S1	17 (25.4)
**Total number of instrumented levels**	67
**Previous non-instrumented surgery ^c^**, n (%)	6 (21.4)

**^a^** indicates how many levels have been instrumented with pedicle screws in different patients. **^b^** indicates which vertebral levels have been instrumented with pedicle screws in different patients. **^c^** indicates how many patients in the study population had previous spinal surgery without instrumentation.

**Table 2 bioengineering-10-00872-t002:** Brantigan–Steffee classification of spinal fusion.

Classification	Description	Number of Cases
A- Obvious radiographic pseudoarthrosis	Pseudoarthrosis, collapse of construct, loss of disc height, vertebral slip, broken screw, displacement of the cage, resorption of bone graft	0
B- Probable pseudoarthrosis	Significant resorption of bone graft, major lucency, or gap visible in the fusion area > 2 mm	0
C- Radiographic status uncertain	A small lucency or gap may be visible with at least half of the graft area showing no lucency between the graft bone and the vertebral bone	4
D- Probable radiographic fusion	Bone bridges the entire fusion area with at least the density originally achieved at surgery. There should be no lucency between the graft bone and the vertebral bone.	4
E- Radiographic fusion	The bone in the fusion area is more dense and more mature than originally achieved at surgery; there is no interface between the donor bone and the vertebral bone: a sclerotic line between the graft bone and the vertebral bone indicates solid fusion. Other indicators of solid fusion are fusion at the facet joints and anterior progression of the graft in the disc.	20

**Table 3 bioengineering-10-00872-t003:** Survey about hardware’s perception: “When do you feel the presence of the implants in your spine?”.

Answer	Number of Patients (%)
1. Never	20
2. Only during weather/temperature change	6
3. Occasionally	1
4. Always	0

One patient did not respond to the survey and PROMs.

**Table 4 bioengineering-10-00872-t004:** Thermal properties of CFR-PEEK and bone.

	CFR-PEEK	Cortical Bone	Trabecular Bone
Coefficients of thermal expansion (×10^−5^ mm/K)	4 ÷ 6 [37] 5.4 ÷ 39 [38]	2.8 ± 0.4 [39]	n.a.
Thermal conductivity (W/m·K)	0.66 [37] 0.95 [38]	0.68 ± 0.01 [40]	0.39 ± 0.06 [41] 0.26 ÷ 0.33 [42]

## Data Availability

Data supporting the reported results can be retrieved by asking to the corresponding authors.

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
