# Peer review of "Carbon Fiber-Reinforced PolyEtherEtherKetone (CFR-PEEK) Instrumentation in Degenerative Disease of Lumbar Spine: A Pilot Study"

_bioengineering, 2023, doi:10.3390/bioengineering10070872_

Round 1

Reviewer 1 Report

the paper with the title "Carbon Fiber Reinforced-PolyEtherEtherKetone (CFR-PEEK) instrumentation in degenerative disease of lumbar spine: a pilot study" was really interesting. minor review in several parts need to be clarified before it goes to be published.

1. Abstract should be consist of a single paragraph. make it compact

2. abstract should be added with the most important results in value mode. i.e. XXMPA or XX%.

3. Please add 2 or 3 more references to strengthen this claim. "In the last decades metal implants have become commonly used in spinal surgery for the stabilization of the column with pedicle screws and rods to treat different pathologie of the spine, and such instrumentation has dramatically increased the union rate in spinal fusion. Due to its strength, biocompatibility and possibility to be contoured intraoperatively for a tailored fit, titanium has become the metal of choice for spine surgery, replacing stainless steel implants [1]."

4. This paragraph is too short and no references that added to claim this statement. “However, metal implants also have disadvantages which include limited fatigue life, mismatch of modulus elasticity between the metal and the bone, potential for debris generation, allergic host response, imaging artifacts in MRI, X-rays and CT evaluation.” Please revised it. It can be added with 1 or 2 sentences or combined it with previous paragraph.

5. The used of carbon fiber are widely used in different application. Start with this term and then continue to give example of the applications and then focus on the biomedical application. Several papers can be used i.e. Recent Progress on Natural Fibers Mixed with CFRP and GFRP: Properties, Characteristics, and Failure Behaviour, Experimental and numerical analysis of CFRP-SPCC hybrid laminates for automotive and structural applications with cost analysis assessment.

6. In the materials and methods, schematic study or step by step from preparation sample up to testing and evaluation should be added in a single figure.

7. this section should be combined with previous paragraph. “These biomechanical properties prompted us to think that the use of CRF-PEEK could be advantageous also for the stabilization of the spine in patients affected by degenerative diseases.”

8. this section also should be combine with previous paragraph. “This paper analyses a case series of patients treated with CRF-PEEK instrumentation for the stabilization of the spine in degenerative lumbar spine disease, investigating the radiographic results in terms of fusion rate and the clinical outcomes.”

9. These 3 paragraphs should be combine.

All the patients enrolled received a standard open PLIF/TLIF procedure with a com- 101 posite CFR-PEEK fixation system (CarboClear or BlackArmor) and a combined PEEK core/titanium surfaced interbody cage (Concorde® ProTi 360° System, Depuy Synthes, USA, with dimensions 9x23 or 9x27 and height ranging from 7 to 15 mm).

All the patients were submitted to post-op CT scan and standard radiograms before discharge. During the follow-up period all the patients had clinical examinations at 6 and 12 months and were evaluated at 12 months with CT scan to assess implant stability and fusion. Patient-reported outcomes (VAS, SF-12 and EQ-5D scores) were also collected at baseline and at 12 months follow up. Adverse events were collected in the intra-operative, post-operative and follow up periods and classified according to SAVES v2 [21].

Moreover, the authors asked to the patients if they felt the presence of spinal instrumentation and if this presence was particularly relevant in association with a change in weather or temperature. The possible answers to the question: “when you feel the presence of the implants in your spine?” were scored as follows: 1 = never; 2 = only during weather/temperature change; 3 = occasionally; 4 = always

10. Fig. 1-3 should be combined.

11. Why only female that just being tested in the present study?

12. Paragraph is consist of 2 or more sentences. Please carefully checked all the entire manuscript if the paragraph only consist of 1 sentence, rather added with other sentences or combined it with previous or next paragraph. This example also need to be combine. “All these papers report about clinical applications of CRF-PEEK systems for the treatment of spinal tumours, where the advantages of radiolucency and artifact-free imaging are very attractive for radiotherapy and for treatment monitoring.”

13. This also need to combine. “These encouraging radiographic outcomes were associated to a significant improvement of clinical outcomes, measured by PROs, including pain reduction and improvement of health-related quality of life.”

14.This also need to combine. “Moreover, the effect of cold weather has been investigated in patients with orthopaedic implants, but this was not related to any specific implant type, material, nor site [32].”

15. conclusion should be in a single paragraph, compact, and showed all results. If the test is many, the form can be bullets or point by point. Revised it.

Author Response

Answers to Reviewer 1 

  1. Abstract should consist of a single paragraph. Make it compact
  2. Abstract should be added with the most important results in value mode. i.e. XXMPA or XX%.

Abstract has been modified as suggested.

  1. Please add 2 or 3 more references to strengthen this claim. "In the last decades metal implants have become commonly used in spinal surgery for the stabilization of the column with pedicle screws and rods to treat different pathologies of the spine, and such instrumentation has dramatically increased the union rate in spinal fusion. Due to its strength, biocompatibility and possibility to be contoured intraoperatively for a tailored fit, titanium has become the metal of choice for spine surgery, replacing stainless steel implants [1]."

Done, two references have been added.

  1. This paragraph is too short and no references that added to claim this statement. “However, metal implants also have disadvantages which include limited fatigue life, mismatch of modulus elasticity between the metal and the bone, potential for debris generation, allergic host response, imaging artifacts in MRI, X-rays and CT evaluation.” Please revised it. It can be added with 1 or 2 sentences or combined it with previous paragraph.

Done

  1. The use of carbon fiber are widely used in different application. Start with this term and then continue to give example of the applications and then focus on the biomedical application. Several papers can be used i.e. Recent Progress on Natural Fibers Mixed with CFRP and GFRP: Properties, Characteristics, and Failure Behaviour, Experimental and numerical analysis of CFRP-SPCC hybrid laminates for automotive and structural applications with cost analysis assessment.

Done, two references have been added.

  1. In the materials and methods, schematic study or step by step from preparation sample up to testing and evaluation should be added in a single figure.

Done, we added a schematic representation of timing and procedures (Figure 1). The Figure has been uploaded ad Pdf file, while it was difficult to embed it in the manuscript.

  1. This section should be combined with previous paragraph. “These biomechanical properties prompted us to think that the use of CRF-PEEK could be advantageous also for the stabilization of the spine in patients affected by degenerative diseases.”

Done

  1. this section also should be combined with previous paragraph. “This paper analyses a case series of patients treated with CRF-PEEK instrumentation for the stabilization of the spine in degenerative lumbar spine disease, investigating the radiographic results in terms of fusion rate and the clinical outcomes.”

Done

  1. These 3 paragraphs should be combined.

All the patients enrolled received a standard open PLIF/TLIF procedure with a com- 101 posite CFR-PEEK fixation system (CarboClear or BlackArmor) and a combined PEEK core/titanium surfaced interbody cage (Concorde® ProTi 360° System, Depuy Synthes, USA, with dimensions 9x23 or 9x27 and height ranging from 7 to 15 mm).

All the patients were submitted to post-op CT scan and standard radiograms before discharge. During the follow-up period all the patients had clinical examinations at 6 and 12 months and were evaluated at 12 months with CT scan to assess implant stability and fusion. Patient-reported outcomes (VAS, SF-12 and EQ-5D scores) were also collected at baseline and at 12 months follow up. Adverse events were collected in the intra-operative, post-operative and follow up periods and classified according to SAVES v2 [21].

Moreover, the authors asked to the patients if they felt the presence of spinal instrumentation and if this presence was particularly relevant in association with a change in weather or temperature. The possible answers to the question: “when you feel the presence of the implants in your spine?” were scored as follows: 1 = never; 2 = only during weather/temperature change; 3 = occasionally; 4 = always

Done

  1. Fig. 1-3 should be combined.

Done, a new Figure 3 now includes four panels indicated with letters a, b, c, d.

  1. Why only female that just being tested in the present study?

As reported in Table 1, female and male were equally represented in our study population. In Figures 4 and 5 we reported pre-operative and post-operative radiographic images of two patients randomly selected in order to show the intervention and the radiographic outcomes, and it is a coincidence that they are two females.

  1. Paragraph is consist of 2 or more sentences. Please carefully checked all the entire manuscript if the paragraph only consist of 1 sentence, rather added with other sentences or combined it with previous or next paragraph. This example also need to be combine. “All these papers report about clinical applications of CRF-PEEK systems for the treatment of spinal tumours, where the advantages of radiolucency and artifact-free imaging are very attractive for radiotherapy and for treatment monitoring.”

Done

  1. This also need to combine. “These encouraging radiographic outcomes were associated to a significant improvement of clinical outcomes, measured by PROs, including pain reduction and improvement of health-related quality of life.”

Done

14.This also need to combine. “Moreover, the effect of cold weather has been investigated in patients with orthopaedic implants, but this was not related to any specific implant type, material, nor site [32].”

Done

  1. Conclusion should be in a single paragraph, compact, and showed all results. If the test is many, the form can be bullets or point by point. Revised it.

We think that Conclusions are properly exposed.

Reviewer 2 Report

This manuscript serves as a pre-study of the CFR-PEEK instrumentation in lumbar spine disease. Some results and discussions have been provided. To improve the manuscript, here are some suggestions:

1. In the abstract, the items of "background", "methods", and "conclusions" are not needed.

2. It is highly suggested to provide images of the implants, screws... and corresponding detailed descriptions. Not every reader of this journal has such background.

3. Similar to comment 2, it is also suggested to provide background information (with images) of the items shown in Table 1. Such as number of instrumented levels, levels instrumented.

4. Figure 3 a&b. the stars (***) should be in one row, right?

5. It is highly recommended to have at least one image with the spine and the implant together with some description.

6. Is there any comparison with CFR-PEEK and metal materials? With mechanical properties, fatigue life, cost, and other related properties.

7. The discussion part provided lots of current studies, which seems more appropriate in the introduction part.

Author Response

Answers to Reviewer 2 

This manuscript serves as a pre-study of the CFR-PEEK instrumentation in lumbar spine disease. Some results and discussions have been provided. To improve the manuscript, here are some suggestions:

  1. In the abstract, the items of "background", "methods", and "conclusions" are not needed.

Items have been deleted.

  1. It is highly suggested to provide images of the implants, screws... and corresponding detailed descriptions. Not every reader of this journal has such background.

As suggested, a new Figure has been added to present the spinal instrumentation into the column (Figure 2).

  1. Similar to comment 2, it is also suggested to provide background information (with images) of the items shown in Table 1. Such as number of instrumented levels, levels instrumented.

Notes have been added in Table 1 to better explain the meaning of instrumented levels.

  1. Figure 3 a&b. the stars (***) should be in one row, right?

We corrected the mistake.

  1. It is highly recommended to have at least one image with the spine and the implant together with some description.

It is described in the new Figure 1.

  1. Is there any comparison with CFR-PEEK and metal materials? With mechanical properties, fatigue life, cost, and other related properties.

Data concerning the characteristics of metal implants and CRF-PEEK implants have been reported in the literature, as described in the manuscript. The comparison between the two materials in terms of mechanical properties and costs was not the aim of our study. A randomized clinical trial has been designed by the Authors in order to further investigate the results obtained in this pilot study.

  1. The discussion part provided lots of current studies, which seems more appropriate in the introduction part.

We think that the introduction and the discussion are fairly balanced. The Introduction describes in general metals and carbon fibers for spinal surgery, while the Discussion focuses on and deepens the relevant literature concerning CRF-PEEK materials used in our study.

Round 2

Reviewer 1 Report

After carefully check the manuscript, the present paper can be accepted.

Reviewer 2 Report

Suggestions have been responded appropriately.